# Speedrunning Tabular Foundation Model Pretraining

**Salih Bora Öztürk** [1]    **Alexander Pfefferle** [2 1]    **Frank Hutter** [3 2 1]

## Abstract

Pretraining cost is a major bottleneck for research on tabular foundation models, slowing the iteration cycle for new architectures, priors, and optimization ideas. Yet the community lacks a simple way to compare and accumulate pretraining speedups. We introduce a community speedrun for nanoTabPFN: contributors modify a single-file training script and compete to reach a fixed downstream ROC AUC target on subsampled TabArena using one NVIDIA L40S GPU. The current best record reaches the target in 0.92 minutes, an $81\times$ speedup over the 74.32-minute baseline while using $22\times$ fewer synthetic datasets. The speedrun format provides a simple protocol for the community to add, verify, and stack pretraining improvements, with the leaderboard open to contributions. Code and records are available at https://github.com/borawhocodess /modded-nanotabpfn.

## 1. Introduction

Structured data in tables is the bedrock of decision making (Borisov et al., 2024; van Breugel & van der Schaar, 2024). Tabular foundation models (TFMs) are fundamentally changing the field, much as LLMs did for text (Hollmann et al., 2023; 2025; Grinsztajn et al., 2026; Qu et al., 2025; 2026; Ma et al., 2025; Zhang et al., 2025). However, pretraining foundation models is computationally expensive, taking hours to days even at small scale and slowing research iteration. Considerable work has tackled the problem of speeding up LLM pretraining (Jordan et al., 2024a; Geiping & Goldstein, 2023; DeepSeek-AI et al., 2024; OpenAI, 2026), but to our knowledge no equivalent effort exists for TFMs. We close this gap with a public speedrun built around nanoTabPFN (Pfefferle et al., 2025), empirically

[1]University of Freiburg, Freiburg im Breisgau, Germany [2]ELLIS Institute Tübingen, Tübingen, Germany [3]Prior Labs, Freiburg im Breisgau, Germany. Correspondence to: Salih Bora Öztürk <oeztuers@cs.uni-freiburg.de>.

*Proceedings of the $2^{nd}$ ICML Workshop on Foundation Models for Structured Data*, Seoul, South Korea. 2026. Copyright 2026 by the author(s).

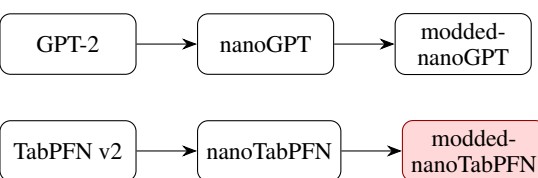

*Figure 1.* Two parallel speedrun lineages. The language side (top) is established. modded-nanoTabPFN (red) is the contribution of this paper.

testing which techniques transfer from language modeling. The current record at the time of writing reduces pretraining wallclock time from our baseline of 74.32 minutes to 0.92 minutes on a single GPU (NVIDIA L40S), an $81\times$ speedup, while matching the baseline predictive performance on subsampled TabArena datasets (Erickson et al., 2025). The contribution is not a new tabular model family, rather it is a fixed and reproducible speedrun protocol and a public sequence of records that isolates which practical pretraining optimizations transfer to TFMs. The remainder of this paper details the rules, evaluation, and the techniques behind the records.

## 2. Related Work

**Tabular Foundation Models.** Tabular foundation models pretrain a transformer, typically on synthetic datasets, and then make predictions on new tasks via in-context learning. TabPFN (Hollmann et al., 2023; 2025; Grinsztajn et al., 2026) pioneered this approach, followed by TabICL (Qu et al., 2025; 2026), TabDPT (Ma et al., 2025), and LimiX (Zhang et al., 2025). All these models share an expensive pretraining stage which limits fast prototyping of research ideas.

**nanoTabPFN.** nanoTabPFN (Pfefferle et al., 2025) is a minimal, educational reimplementation of TabPFN v2 and the basis for our competition. While nanoTabPFN's fast pretraining came from training and evaluating on very small datasets, our baseline scales up the architecture and the size of the datasets we pretrain on. We also significantly increase the size limits of the datasets we evaluate on ($5\times$ more datapoints and $10\times$ more features), making it harder to reach the target performance and thereby significantly increasing the baseline training time. Additionally we also evaluate on

all classification tasks of TabArena by subsampling both features and datapoints if they are outside limits, where as nanoTabPFN originally subsampled datapoints but filtered out datasets that exceeded the feature limit. The most important difference between nanoTabPFN and modded-nanoTabPFN is that the former focused on having very simple and easy-to-understand code for educational purposes, where as in modded-nanoTabPFN we are interested in any techniques that speed up training, even if they require complex, hard-to-understand implementations.

**nanoGPT and modded-nanogpt.** Our format is inspired by community speedrun efforts on language model pretraining. nanoGPT (Karpathy, 2023) is a minimal, hackable reimplementation of GPT-2 designed for easy prototyping and experimentation. modded-nanogpt (Jordan et al., 2024a) is a community-driven speedrun built on top of nanoGPT in which contributors compete to reach a fixed validation loss in less wallclock time. Iterations of the competition have surfaced techniques such as the Muon optimizer (Jordan et al., 2024b). We adopt the same speedrun leaderboard format for tabular foundation models.

## 3. Competition

Our competition is an open community speedrun, in which contributors compete to reach a fixed downstream-accuracy target on fixed hardware in the shortest wallclock time. Following modded-nanogpt, contributors modify a single training script, which logs its own source code, configuration, software versions, GPU metadata, peak memory, and timing information so every record can be reconstructed from the submitted logs. The official wallclock budget is cumulative training time only, excluding evaluation and prior generation.

**Goal.** The goal is to pretrain a neural network that beats the average validation ROC AUC of a Random Forest baseline on subsampled TabArena datasets, using 1 NVIDIA L40S, in the shortest wallclock time. The target is derived from the same evaluation pipeline as the speedrun model (Appendix A.3).

**Rules.** The rules are minimal and exist only to keep records comparable. A new record must not change the evaluation pipeline, must not load pretrained weights, and must run faster than the prior record when re-run on the verifier's hardware at the same seed. To suppress cluster noise, the reported wallclock is the median of multiple verification runs (Appendix B), and each submitted log carries the metadata needed to reconstruct the run (Appendix E). Beyond that, anything is fair game.

| # | Time (min) | Datasets | Technique |
|---|-----------|----------|-----------|
| 1 | 74.32 | 80,576 | Baseline |
| 2 | 54.41 | 45,824 | Muon optimizer |
| 3 | 10.10 | 13,184 | SDPA, bf16, LR, width |
| 4 | 9.26 | 13,184 | Batched Muon, compile |
| 5 | 7.57 | 11,200 | Residual decay |
| 6 | 3.88 | 9,664 | RMSNorm, Thinking Rows |
| 7 | 3.48 | 8,768 | LAWA, AdamW WD |
| 8 | 2.15 | 4,992 | Repeated feature grouping |
| 9 | 0.92 | 3,648 | HPO, Muon WD, Mean-pool |

*Table 1.* Speedrun records table

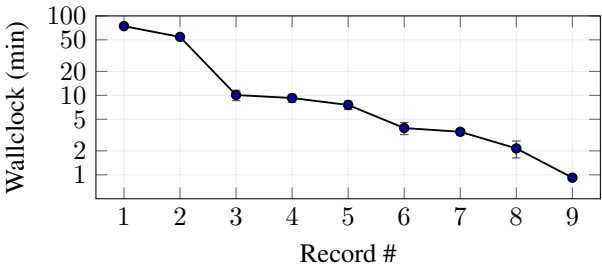

*(a)* Wallclock to target.

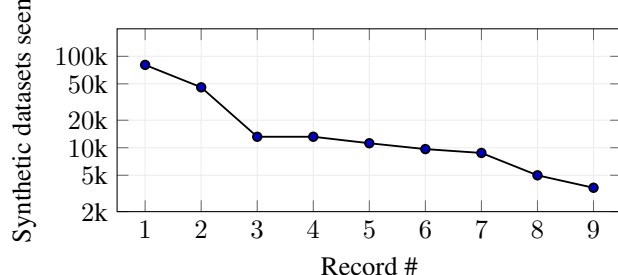

*(b)* Synthetic datasets to target.

*Figure 2.* Records of Table 1 on log scale.

**Evaluation.** Evaluation is on the 38 classification tasks of TabArena (Erickson et al., 2025), a benchmark for tabular machine learning that curates a representative collection of classification and regression tasks. We do not adopt the full benchmark protocol but use the curated datasets only, subsampled per task to at most 100 features and 1000 rows for fast iteration. This lets us inherit the dataset diversity while keeping evaluation fast enough for rapid iteration. For each task we run 5-fold stratified cross-validation with per-fold preprocessing fitted on train only, concatenate out-of-fold predictions across folds, and score once with ROC AUC, binary for two-class tasks and one-vs-rest for multiclass. We average the per-task scores over the 38 tasks and run this evaluation periodically during training, stopping as soon as the target (performance equivalent to a random forest) is reached. The full pipeline and the task list are in Appendices A.2 and A.1.

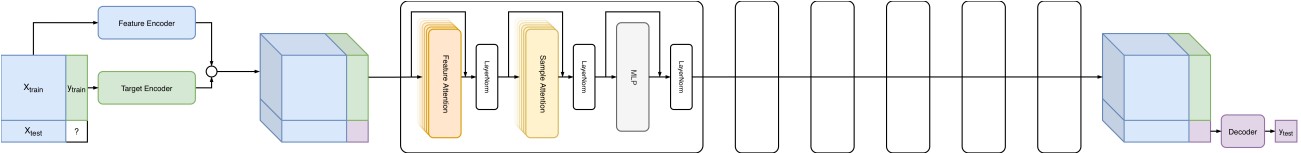

*Figure 3.* High-level architecture of the baseline. Input features $x$ are encoded with a feature encoder, target labels $y$ with a target encoder and extended with the train-row mean to match the data shape. The two streams are merged and passed through a stack of 6 transformer blocks, each applying feature-axis attention, norm, sample-axis attention, norm, MLP, norm. The test-row outputs are sliced out and fed into the decoder.

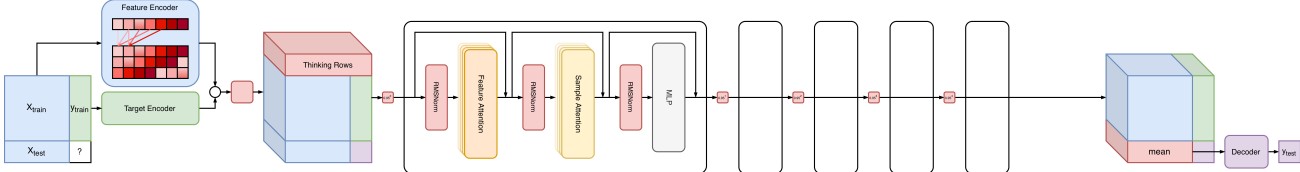

*Figure 4.* High-level architecture of the current best record. Changes from the baseline are marked in red. Input features are grouped at the feature encoder, 24 learnable thinking rows are appended along the data axis, the residual stream entering each block is decayed by $0.95^i$, the norms move ahead of attention and MLP, and the stack is reduced to 5 blocks. The mean over feature embeddings at the test rows is fed into the decoder instead of the target slice.

## 4. Speedrun Results

The records collected so far compress pretraining from 74.32 minutes on the baseline to 0.92 minutes on the current best, an $81\times$ wallclock speedup. The synthetic dataset count drops from 80,576 to 3,648 over the same interval, as summarized in Table 1 and Figure 2. Each subsection below walks through one record, naming the techniques it introduced and its resulting wallclock time. Each subsequent record builds on the previous one. See Appendix C for a side-by-side configuration of the baseline against the current best, and Appendix B for per-record details and statistics.

### 4.1. Baseline

The baseline is nanoTabPFN consolidated into a single training script: a 6-layer transformer encoder with 6 heads, embedding size 192, and MLP hidden size 768, using post-norm LayerNorms (Ba et al., 2016) and trained with schedule-free AdamW (Defazio et al., 2024), batch size 1, with evaluation every 64 steps, in fp32 throughout. The high-level inference architecture is shown in Figure 3. The pretraining prior is a static dump of synthetic classification datasets, each with 1,000 rows, up to 20 features, and up to 8 classes, generated by the TabICL prior (Qu et al., 2025). It reaches the target after 80,576 synthetic datasets and 74.32 minutes on a single L40S, anchoring the start of the speedrun.

### 4.2. Muon Optimizer

The first record uses the Muon optimizer (Jordan et al., 2024b) for the 2D weight matrices of the transformer encoder, while keeping the rest with the baseline's schedule-free AdamW. Previously validated in image classification and language-model pretraining, Muon transfers to tabular foundation models with the same hyperparameters as in modded-nanogpt and a Muon learning rate set to $0.1\times$ the schedule-free AdamW learning rate. Wallclock time drops from 74.32 to 54.41 minutes and the synthetic dataset count from 80,576 to 45,824, showing sample efficiency on tabular pretraining as well.

### 4.3. SDPA, bf16, Learning Rate, Width

The next record bundles a scaled dot-product attention (SDPA) rewrite, a switch from post- to pre-Norm in the transformer blocks (Xiong et al., 2020), and mixed precision training with hyperparameter tunings: a learning rate increase to $10^{-3}$ and a wider embedding ($192 \rightarrow 256$) with fewer heads ($6 \rightarrow 4$). Ablations isolate the learning-rate increase and the SDPA rewrite as the dominant drivers, with bf16 and TF32 matmul contributing the remainder. Wallclock time drops from 54.41 to 10.10 minutes and the synthetic dataset count from 45,824 to 13,184 (per-component ablation in Appendix B).

### 4.4. Batched Muon, Compile

The next record batches the Newton–Schulz iteration across the QKV weight matrices and compiles the transformer encoder layer's forward pass. Both are pure throughput optimizations: wallclock drops from 10.10 to 9.26 minutes while the synthetic dataset count is unchanged at 13,184.

### 4.5. Residual Decay

The next record scales the input to each transformer block by an exponentially-decaying factor, with layer $i$'s input scaled by $0.95^i$, exponentially down-weighting earlier-layer contributions in the final output. Wallclock drops from 9.26 to 7.57 minutes and the synthetic dataset count from 13,184 to 11,200.

### 4.6. RMSNorm, Thinking Rows

The next record replaces LayerNorms with lower-precision RMSNorm (Zhang & Sennrich, 2019) and prepends 16 learnable thinking rows (Grinsztajn et al., 2026) to the input. The data rows attend to these alongside each other in row-axis attention, and the thinking rows account for most of the improvement. The trained attention maps confirm the thinking rows are actively attended to. Per-component ablation and the attention maps are in Appendix B.6. Wallclock drops from 7.57 to 3.88 minutes and the synthetic dataset count from 11,200 to 9,664.

### 4.7. LAWA, AdamW Weight Decay

The next record adds Latest Weight Averaging (Kaddour, 2022; Sanyal et al., 2024), which maintains a FIFO of the last 10 checkpoints and averages them into a temporary model for evaluation only. Also a weight decay of 0.01 is applied to the schedule-free AdamW parameters. Wallclock drops from 3.88 to 3.48 minutes and the synthetic dataset count from 9,664 to 8,768.

### 4.8. Repeated Feature Grouping

The next record adopts repeated feature grouping from TabI-CLv2 (Qu et al., 2026): column $j$ is embedded jointly with columns $j+1$ and $j+3$ (mod $m$), so every column participates in three overlapping groups, addressing representation collapse when features share similar distributions. Wallclock drops from 3.48 to 2.15 minutes and the synthetic dataset count from 8,768 to 4,992.

### 4.9. Autoresearch HPO, Muon Weight Decay, Mean-pool Decoder

The current best record was discovered by adapting autoresearch (Karpathy, 2026), an LLM-driven hyperparameter and architecture search loop, over multiple runs with human intervention. The changes comprise hyperparameter tuning, weight decay added to Muon, and feeding the decoder the mean over feature embeddings of test rows instead of the target embeddings of test rows. Wallclock drops from 2.15 to 0.92 minutes and the synthetic dataset count from 4,992 to 3,648.

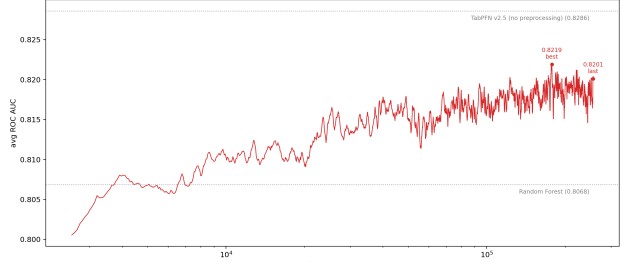

*Figure 5.* Long-run training trajectory of modded-nanoTabPFN. The x-axis is on a log scale and starts at 2,560 synthetic datasets for visual clarity. Markers indicate the best and last values reached. Dotted lines show the Random Forest target where the speedrun normally stops, and TabPFN v2.5 with no preprocessing as an upper reference.

## 5. Beyond the Target

The high-level inference architecture of the current best record is shown in Figure 4. While the records optimize pretraining wallclock time to a fixed target, a complementary question is what this model achieves if trained for the baseline's full budget. Trained until the 256,000-dataset prior dump is exhausted (in 70.88 minutes), modded-nanoTabPFN reaches an average ROC AUC of approximately 0.82 on subsampled TabArena, approaching TabPFN v2.5 at 0.8286 with preprocessing and ensembles disabled (Figure 5). At around the same wallclock time the baseline is at 0.8066 after 76,544 synthetic datasets, essentially still at the Random Forest target, while ours see the entire 256,000-dataset prior in the same wallclock budget. Long-run setup details are in Appendix D.

## 6. Conclusion

We introduced an open speedrun for tabular foundation model pretraining built around nanoTabPFN, with a fixed downstream target on subsampled TabArena datasets. The current best record reaches the Random Forest target in 0.92 minutes on a single L40S, $81\times$ faster than the 74.32-minute baseline, and using $22\times$ fewer synthetic datasets. The same recipe scaled to a longer budget keeps improving past the speedrun target, suggesting pretraining ideas selected under tight wallclock also pay off when the budget is relaxed.

Beyond the numbers, the speedrun itself is the contribution, providing a simple protocol for the community to add, verify, and stack pretraining improvements. Every record so far holds the synthetic prior fixed, making the prior itself the most promising untouched direction. The leaderboard is open to the tabular foundation model community, and contributions are welcome.

## Acknowledgements

Funded by the European Union. Views and opinions expressed are however those of the author(s) only and do not necessarily reflect those of the European Union or the European Commission. Neither the European Union nor the European Commission can be held responsible for them. This work was supported by the European Union's Horizon Europe research and innovation programme under grant agreement No 101214398 (ELLIOT).

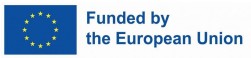 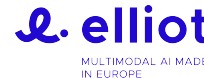

Frank Hutter acknowledges the financial support of the Hector Foundation. We thank the reviewers for their feedback and Carter Prince for his contributions to the records. We are also grateful to the PyTorch (Paszke et al., 2019) contributors.

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

# A. Evaluation Details

## A.1. TabArena Tasks

Table 2 lists the 38 TabArena classification tasks used for evaluation. For subsampling details see Appendix A.2.

*Table 2.* The 38 TabArena classification tasks. Row and feature counts are the original task sizes before subsampling.

| # | Task ID | Rows | Features | Name |
|---|---------|------|----------|------|
| 1 | 363613 | 32,769 | 10 | Amazon_employee_access |
| 2 | 363614 | 898 | 39 | anneal |
| 3 | 363616 | 76,000 | 171 | APSFailure |
| 4 | 363618 | 45,211 | 14 | bank-marketing |
| 5 | 363619 | 10,000 | 11 | Bank_Customer_Churn |
| 6 | 363620 | 3,751 | 1,777 | Bioresponse |
| 7 | 363621 | 748 | 5 | blood-transfusion-service-center |
| 8 | 363623 | 5,000 | 20 | churn |
| 9 | 363624 | 9,822 | 86 | coil2000_insurance_policies |
| 10 | 363626 | 1,000 | 21 | credit-g |
| 11 | 363627 | 30,000 | 24 | credit_card_clients_default |
| 12 | 363628 | 129,880 | 22 | customer_satisfaction_in_airline |
| 13 | 363629 | 768 | 9 | diabetes |
| 14 | 363630 | 71,518 | 48 | Diabetes130US |
| 15 | 363632 | 10,999 | 11 | E-CommereShippingData |
| 16 | 363671 | 1,500 | 7 | Fitness_Club |
| 17 | 363673 | 150,000 | 11 | GiveMeSomeCredit |
| 18 | 363674 | 2,400 | 31 | hazelnut-spread-contaminant-detection |
| 19 | 363676 | 10,459 | 24 | heloc |
| 20 | 363677 | 3,845 | 1,618 | hiva_agnostic |
| 21 | 363679 | 19,158 | 13 | HR_Analytics_Job_Change_of_Data_Scientists |
| 22 | 363681 | 12,684 | 25 | in_vehicle_coupon_recommendation |
| 23 | 363682 | 1,723 | 14 | Is-this-a-good-customer |
| 24 | 363683 | 50,000 | 213 | kddcup09_appetency |
| 25 | 363684 | 2,240 | 26 | Marketing_Campaign |
| 26 | 363685 | 1,014 | 7 | maternal_health_risk |
| 27 | 363689 | 7,491 | 87 | NATICUSdroid |
| 28 | 363691 | 12,330 | 18 | online_shoppers_intention |
| 29 | 363694 | 5,910 | 65 | polish_companies_bankruptcy |
| 30 | 363696 | 1,054 | 42 | qsar-biodeg |
| 31 | 363699 | 78,053 | 12 | SDSS17 |
| 32 | 363700 | 2,584 | 16 | seismic-bumps |
| 33 | 363702 | 3,190 | 61 | splice |
| 34 | 363704 | 4,424 | 37 | students_dropout_and_academic_success |
| 35 | 363706 | 6,819 | 95 | taiwanese_bankruptcy_prediction |
| 36 | 363707 | 1,353 | 10 | website_phishing |
| 37 | 363711 | 1,699 | 112 | MIC |
| 38 | 363712 | 10,885 | 22 | jm1 |

## A.2. Evaluation Pipeline

- **Subsampling.** If a task has more than 100 features, 100 are selected uniformly at random. If a task has more than 1000 samples, 1000 are selected stratified by class label. Each task is subsampled with a fixed random seed for reproducibility.

- **Cross-validation.** 5-fold `StratifiedKFold` with shuffling. Class labels are integer-encoded per fold by a `LabelEncoder` fitted on the training labels.

- **Preprocessing.** Applied per fold, fitted on training data only:
  - Columns with at most one unique non-NaN value are dropped.
  - Numeric columns: `pd.to_numeric` coercion followed by mean imputation.
  - Categorical columns: ordinal encoding (unknown values map to NaN) followed by most-frequent imputation.

- **Metric.** For each task, out-of-fold predictions are concatenated across the 5 folds and scored once with `roc_auc_score`: binary ROC AUC for two-class tasks and one-vs-rest ROC AUC for multi-class tasks. The reported score is the mean of the 38 per-task ROC AUCs.

All sources of randomness (feature subsampling, row subsampling, fold splits) are seeded with the same seed (11).

### A.3. Random Forest Target

The competition's downstream target $0.8068462330697953$ is the average validation ROC AUC of an off-the-shelf scikit-learn `RandomForestClassifier` run through the same evaluation pipeline as the speedrun model. The number is hardcoded as the `jackpot` in the training script so every record runs against the same target.

## B. Record Details

The training script logs its own source code, so each record is captured by a single log file. We verify each record with multiple cluster runs to determine its reported time. Cluster hosts vary in speed, and the changes from Record 3 onward make training numerically non-deterministic, so both wallclock pretraining time and dataset count vary across runs. We report the median, which is robust to slow node outliers and selects the verification run whose log is submitted as the record. The baseline (Record 1) is the only exception: we report its mean, since it sets the reference time the rest of the records are compared against. Per-record verification statistics are summarized in Table 3; subsections below add details specific to individual records.

*Table 3.* Verification statistics per record. Mean, standard deviation, and median are wallclock pretraining times in minutes. "Datasets" is the synthetic-dataset count at the median run.

| # | Record | Runs | Mean | Std | Median | Datasets |
|---|--------|------|------|-----|--------|----------|
| 1 | Baseline | 5 | 74.32 | 0.90 | 74.60 | 80,576 |
| 2 | Muon Optimizer | 5 | 54.49 | 0.29 | 54.41 | 45,824 |
| 3 | SDPA, bf16, LR, Width | 5 | 10.56 | 1.53 | 10.10 | 13,184 |
| 4 | Batched Muon, Compile | 5 | 9.29 | 1.07 | 9.26 | 13,184 |
| 5 | Residual Decay | 31 | 7.43 | 0.93 | 7.57 | 11,200 |
| 6 | RMSNorm, Thinking Rows | 23 | 4.05 | 0.68 | 3.88 | 9,664 |
| 7 | LAWA, AdamW Weight Decay | 27 | 3.48 | 0.30 | 3.48 | 8,768 |
| 8 | Repeated Feature Grouping | 31 | 2.28 | 0.52 | 2.15 | 4,992 |
| 9 | Autoresearch HPO, Muon weight decay, Mean-pool Decoder | 31 | 0.93 | 0.04 | 0.92 | 3,648 |

### B.1. Baseline

The baseline is the nanoTabPFN-style architecture consolidated into the single-file speedrun script. It uses the static 256,000-dataset TabICL-prior dump but stops as soon as the Random Forest target is met. Evaluation is run every 64 optimizer steps, and the official reported time excludes the evaluation calls themselves. This record is treated as the reference point rather than as a competitor-submitted improvement; consequently Table 3 reports its mean wallclock time in the main records table, while also listing the median for completeness.

### B.2. Muon Optimizer

The implementation is adapted from modded-nanogpt (Jordan et al., 2024a). Muon is applied only to two-dimensional hidden-layer weight matrices in the transformer encoder, while schedule-free AdamW remains responsible for the remaining parameters. This record is primarily a sample-efficiency improvement.

### B.3. SDPA, bf16, LR, Width

We isolate each component by (i) adding it alone to the Muon record and (ii) removing it alone from the bundled record. Times are the median of 5 runs.

| Component | Muon + component | | Record − component | |
|---|---|---|---|---|
| | median | Δ | median | Δ |
| Learning rate $10^{-4} \rightarrow 10^{-3}$ | 31.52 | −42% | 32.62 | +223% |
| Embedding $(192, 6) \rightarrow (256, 4)$ | 45.78 | −16% | 15.49 | +53% |
| Explicit-QKV SDPA | 38.99 | −28% | 22.29 | +121% |
| bf16 autocast (training) | 49.31 | −9% | 10.55 | +4% |
| bf16 autocast (inference) | 49.13 | −10% | 8.87 | −12% |
| TF32 matmul | 45.06 | −17% | 9.49 | −6% |

The learning-rate increase and the SDPA rewrite both are the main drivers.

## B.4. Batched Muon, Compile

This record is intended as a pure throughput change. The Newton–Schulz iteration used by Muon is batched across the QKV matrices, reducing optimizer overhead without changing the model's training objective or evaluation interface. In addition, the transformer encoder layer forward pass is compiled.

## B.5. Residual Decay

A later re-run of the same code over 41 cluster runs reached a median of 6.06 minutes (11,008 datasets at the median run), illustrating cluster-noise variance in the record-time number.

## B.6. RMSNorm, Thinking Rows

Both changes are adopted from the TabPFN v2.6 release file. RMSNorm replaces all three LayerNorms in the encoder block while skipping the FP32 autocast upcast that PyTorch's LayerNorm applies by default. Thinking rows, originally introduced in the TabPFN v2.5 model report, prepend 16 learnable embeddings of dimension $e$ along the row dimension. Each embedding is broadcast across all columns, and the rows count as part of the train portion of the input.

We tested each change in isolation:

| Run | Median time (min) | Median datasets |
|---|---|---|
| RMSNorm alone (11 runs) | 6.32 | 11,264 |
| Thinking rows alone (9 runs) | 4.22 | 10,304 |
| Combined (23 runs) | 3.88 | 9,664 |

Thinking rows alone account for most of the combined improvement; RMSNorm contributes a smaller but real additional gain.

## B.7. LAWA, AdamW Weight Decay

Latest Weight Averaging keeps a FIFO buffer of the last 10 checkpoints. At evaluation time only, the buffered weights are averaged into a temporary model and scored, the training weights themselves continue from the unaveraged trajectory. The record also adds weight decay 0.01 to the schedule-free AdamW parameter group.

## B.8. Repeated Feature Grouping

For a table with $m$ columns, the $j$-th group contains columns at positions $(j, j+1, j+3) \bmod m$. The shift pattern $(0, 1, 3)$ is chosen so that the differences between any two of its elements are all distinct: $1, 2, 3$. This guarantees that for any table with $\geq 7$ columns, no pair of columns co-occurs in more than one group. The implementation is adapted from nanotabicl.

## B.9. Autoresearch

The hyperparameter changes adopted from the autoresearch search loop are listed in Table 4. In addition, weight decay is added to the Muon parameter update, and the decoder is fed the mean over feature tokens at the test rows instead of the target token. We do not claim that every component in this bundle is independently beneficial, a full leave-one-out ablation of the final record is future work.

*Table 4.* Hyperparameter changes adopted from the autoresearch runs.

| Hyperparameter | Before | After |
|---|---|---|
| `batch_size` | 1 | 2 |
| `steps` | 64 | 32 |
| `l` (encoder layers) | 6 | 5 |
| `thinking_rows` | 16 | 24 |
| `feature_group_size` | 3 | 5 |
| `muon_momentum` | 0.95 | 0.96 |
| `grad_clip` | 1.0 | 2.0 |

## C. Baseline and Current Best Configuration

Table 5 summarizes the architecture and training settings of the baseline (Record 1) against those of the current best (Record 9). Empty cells indicate the option does not exist in that configuration.

*Table 5.* Architecture and training configuration of the baseline against the current best record.

| | Baseline | Current Best |
|---|---|---|
| *Architecture* | | |
| Layers ($l$) | 6 | 5 |
| Heads ($a$) | 6 | 4 |
| Embedding ($e$) | 192 | 256 |
| MLP hidden ($h$) | 768 | 768 |
| Norm | post-LayerNorm | pre-LowerPrecisionRMSNorm |
| Feature group size | 1 | 5 |
| Thinking rows | 0 | 24 |
| Residual decay | 1.0 | 0.95 |
| Decoder input | target token | mean of feature tokens |
| *Optimization* | | |
| Optimizer | schedule-free AdamW | schedule-free AdamW + Muon |
| Learning rate | $10^{-4}$ | $10^{-3}$ |
| AdamW weight decay | 0 | 0.01 |
| Muon learning rate | – | $0.1\times$ AdamW |
| Muon momentum | – | 0.96 |
| Muon weight decay | – | 0.1 |
| Gradient clip | 1.0 | 2.0 |
| Latest weight averaging | – | $K=10$ |
| *Throughput* | | |
| Batch size | 1 | 2 |
| Training precision | fp32 | bf16 autocast |

## D. Long-Run Experiment

The long-run training trajectory in Figure 5 is produced by running the current best record's configuration against the same prior dump but with the target effectively disabled so the run does not exit on a target hit. The run was carried out on a single L40S and finished in 4252.72 seconds (70.88 minutes); the highest evaluated mean ROC AUC was 0.8219, with the per-epoch trajectory plotted in Figure 5.

The TabPFN v2.5 reference 0.8286 in the same figure is the score of `TabPFNClassifier` with preprocessing and ensembles disabled (`PreprocessorConfig(name="none")` and `n_estimators=1`) run through the identical evaluation pipeline of Appendix A.2 on the same 38 tasks; preprocessing and ensembling are disabled to compare model capacity at the same evaluation interface.

## E. Reproducibility Metadata

Each submitted record log contains the complete training script, command-line configuration, random seed, target value, evaluation, measured training wallclock, peak CUDA memory, and the host/GPU/software metadata printed at the beginning and end of the run. For hardware comparability, records are verified on a single NVIDIA L40S GPU and are re-run by the verifier rather than accepted from submitter-reported timings.

## F. Use of Generative AI Tools

LLMs were used to set up this speedrun and to iterate on the records, and the current best record was discovered by running autoresearch agents, an LLM-driven agentic loop. Generative AI tools were also used to assist in drafting and editing this manuscript, with all technical content, claims, and experimental results authored and verified by the human authors, who take full responsibility for the contents.

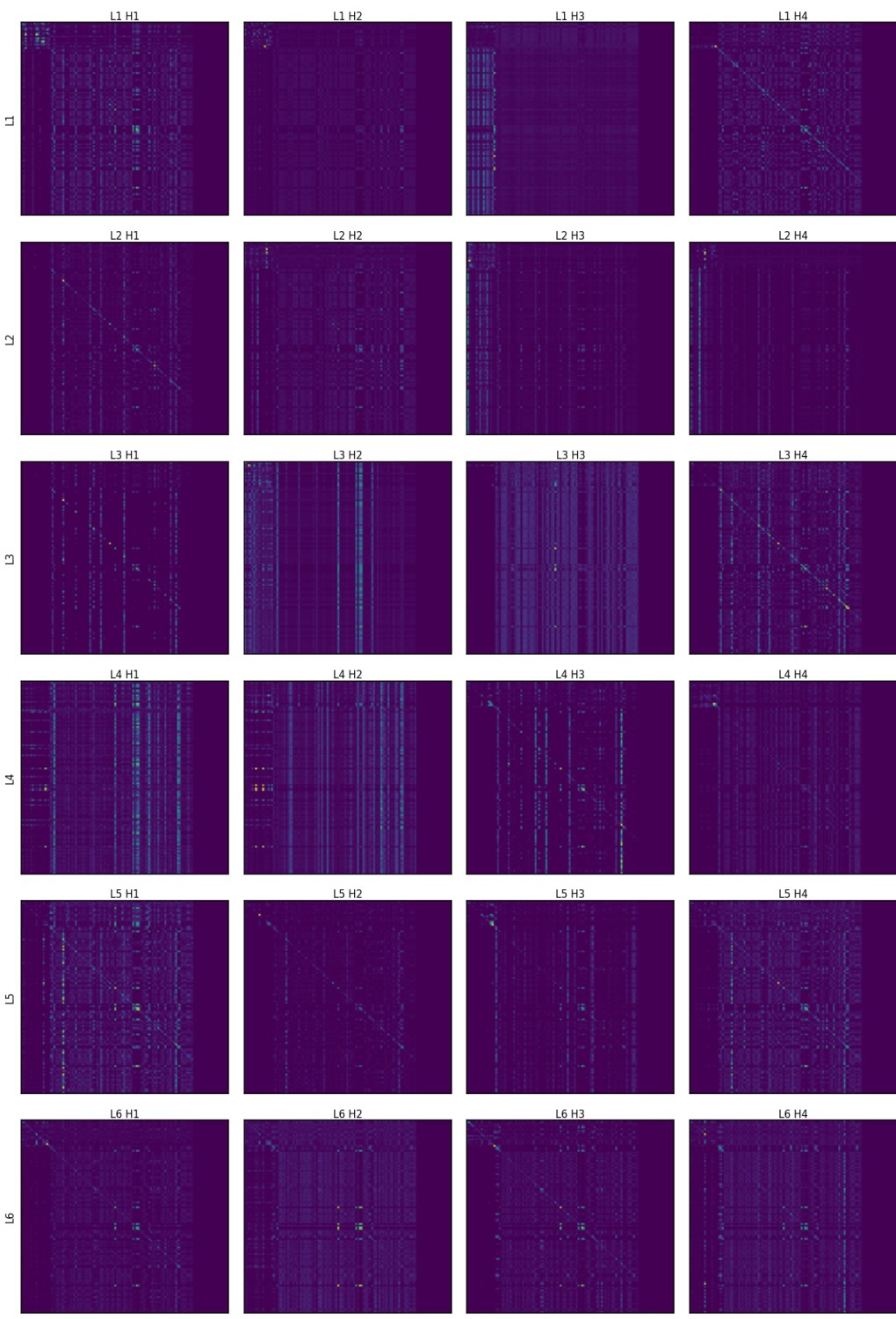

260328-011343-4ba701de-rmsthink-s11-ckpt
rows: 16 think + 80 train + 20 test = 116
white lines mark region boundaries

*Figure 6.* Row attention map of the trained model on the TabArena task `jm1`, subsampled to 100 datapoints for visual clarity, with all encoder layers and heads shown. The first 16 positions on each axis are the prepended thinking rows. In each panel, rows are queries and columns are keys. Vertical bands at the thinking-token columns show data rows placing attention on the thinking tokens, with layer 1 head 3 in particular concentrating most of its mass on those columns.

