# OpenReview forum: "Speedrunning Tabular Foundation Model Pretraining"
_ICML.cc/2026/Workshop/FMSD — FMSD @ ICML 2026 Poster_

### Official Review · Reviewer_LPSc · 2026-05-13
**A Practical and Reproducible Framework for Tabular Foundation Model Pretraining**

**Rating:** 6
**Confidence:** 3

**Review:**

## Summary
This paper introduces a community-driven “speedrun” framework for accelerating tabular foundation model pretraining, built around a lightweight reimplementation of nanoTabPFN and a fixed evaluation target on TabArena classification tasks. By combining a series of optimization techniques and enforcing a reproducible verification protocol, the authors reduce time-to-target.

## Strengths
- The paper addresses an important practical bottleneck in TFM research: slow and expensive pretraining.
- The speedrun setup is simple, transparent, and reproducible, with fixed hardware, verification runs, and public logging.
- The work successfully transfers several effective LLM-training optimizations to the tabular setting.
- Experimental reporting is thorough, including per-record breakdowns, ablations, and long-run training analysis.
- The resulting speedup is substantial and likely valuable for the broader community.

## Weaknesses
- Most improvements are engineering integrations of existing techniques rather than fundamentally new algorithmic ideas.
- The evaluation target is relatively narrow, focusing only on Random Forest parity on subsampled classification tasks.
- The final bundled recipe lacks a full leave-one-out ablation to isolate the contribution of each component.
- Robustness across seeds, priors, and regression settings is not explored sufficiently.
- Some implementation details, such as the residual decay schedule, could be clarified more carefully.

## Overall Assessment
I found this paper well executed and practically useful. While the contribution is primarily infrastructural and empirical rather than algorithmically novel, the community value is significant. The proposed speedrun framework provides a clean and reproducible benchmark for rapid iteration in TFM pretraining, and the reported acceleration results are genuinely impressive. I believe this work can become a useful reference point for future research on efficient tabular pretraining.

---

### Official Review · Reviewer_mHif · 2026-05-21
**A useful infrastructure contribution, weakened by a low accuracy target and a bundled final record that limits attribution.**

**Rating:** 7
**Confidence:** 4

**Review:**

## *Summary*

A community speedrun for nanoTabPFN pretraining: contributors modify a single-file training script and compete to reach a fixed ROC AUC target on subsampled TabArena classification using one NVIDIA L40S in the shortest wallclock time. The target is the average score of an off-the-shelf Random Forest (0.8068) through the same evaluation pipeline. The protocol is modeled on modded-nanogpt. Nine records are documented, taking the baseline from 74.32 min and 80,576 synthetic datasets to 0.92 min and 3,648 datasets, an 81× wallclock and 22× sample-efficiency improvement.

Section 5 runs the current-best configuration to a 70.88-minute long-budget extension and reaches ROC AUC ≈ 0.82, approaching TabPFN v2.5 at 0.8286 (with preprocessing and ensembles disabled). Code and per-record logs are released.

## *Strengths*

- The infrastructure contribution is real. Modded-nanogpt showed that a public, verified, single-file speedrun benchmark can change how a community iterates, and pretraining iteration cost is a current bottleneck in the space. Setting up the TFM analog at this point in the cycle is timely.
- Verification discipline is rigorous. Each record reports median wallclock over 5–31 runs, the script logs its own source code, and records are re-run by the verifier rather than accepted from submitter timings.
- Per-record ablations are presented cleanly. Appendix B.3 isolates each component of the SDPA + bf16 + LR + width bundle and identifies LR and SDPA as the dominant drivers. The thinking-rows vs RMSNorm split in B.6 follows the same protocol.
- The long-run extension in Section 5 is the most important experiment in the paper in my opinion. It shows the recipe selected under tight wallclock keeps paying off at a longer budget, which addresses the natural concern that speedruns overfit to early-training dynamics.
- Scoping is honest as well. Section 4.9 states "we do not claim that every component in this bundle is independently beneficial"; Section 6 flags the synthetic prior as the most promising untouched direction.

## *Areas for Improvement*

- The Random Forest target (0.8068) is a low bar. TabPFN v2.5 reaches 0.8286 on the same pipeline, and the long-run model itself approaches 0.82. As a result, the 81× headline currently reads more as ‘fast to a modest target’ than ‘fast to SOTA. A second tier at the TabPFN v2.5 score would close that gap and make the leaderboard more durable.
- Evaluation is heavily subsampled (100 features, 1,000 rows per task max). Several TabArena tasks have native sizes one to two orders of magnitude larger. It is not entirely clear whether optimizations that win at 1,000-row evaluation transfer to TabArena-native scale, where attention costs and prior-coverage trade-offs change.
- Record 9 carries roughly half the cumulative speedup past Record 6 and bundles three changes (autoresearch HPO, Muon weight decay, mean-pool decoder) with no leave-one-out ablation. The paper acknowledges this is future work, but it is the headline record.
- Single hardware. SDPA, bf16, and compile gains depend on memory bandwidth and tensor-core throughput specific to the L40S, and the record ordering may not transfer to A100 or H100. The protocol should ideally flag this.
- The synthetic prior is held fixed (TabICL prior) throughout. Speedups may be prior-coupled rather than prior-independent. A single prior-swap re-run of one record would test the dependency.
- Contamination is not discussed. The TabArena evaluation comes from OpenML, and the TabICL prior was designed to cover OpenML-like distributions. Whether the 38-task evaluation is operationally independent of the pretraining prior is unclear.
- Classification only. TabArena includes regression; the speedrun does not. Whether techniques like the mean-pool decoder transfer to regression is open and worth pursuing.

## *Detailed Comments*

1. Record 3's clean component ablation in B.3 is the template. Apply the same protocol to Record 9 retrospectively, even as future work.
2. Appendix B.6 shows thinking rows alone at 4.22 min vs combined at 3.88 min, so most of Record 6's gain is the thinking rows rather than RMSNorm. Worth mentioning in Section 4.6.
3. The residual decay factor (0.95^i) appears inherited from modded-nanogpt-adjacent work. A brief sensitivity sweep would establish whether the value is robust or whether the speedup hinges on it.
4. Repeated feature grouping uses shift pattern (0, 1, 3) so pairwise differences are distinct. Worth a sentence on how this generalizes when autoresearch (Record 9) moves group size from 3 to 5.
5. Section B.5 reports a later re-run of Record 5 reaching 6.06 min over 41 runs, against 7.57 min in the main table. Honest, but it suggests the median-over-N protocol may need more runs to stabilize on records where consecutive gaps are under 1 minute.
6. The TabPFN v2.5 reference in Figure 5 disables preprocessing and ensembles to compare model capacity at the same interface. Fair as written, but TabPFN v2.5 with preprocessing and ensembles enabled is the deployment-relevant number, and the gap should be stated.

## *Justification of Score*

The infrastructure is the contribution, and it is well-executed. The protocol (median-of-N verification, log-everything, re-run-by-verifier, single-file scripts) is calibrated for what the paper is trying to convey, and the long-run extension addresses the most obvious concern about speedrun benchmarks. Modded-nanogpt has shown this format can shift how a community iterates; getting the TFM analog into place at this point is timely and fits the FMSD 2026 theme directly.

Two issues hold the score back. The Random Forest target is low enough that the 81× headline reads as "fast to a relatively modest target" rather than "fast to SOTA," and a second tier at the TabPFN v2.5 score would close that. And Record 9, which contributes the largest single jump in cumulative speedup, is also the least experimented and was discovered by an LLM-driven HPO loop, so the human-interpretable takeaway from the headline record is the thinnest one in the table.